# Interventional embolization combined with surgical resection for treatment of extracranial AVM of the head and neck: A monocentric retrospective analysis

**Daniel Lilje**[1], **Martin Wiesmann**[1], **Dimah Hasan**[1◉], **Hani Ridwan**[1◉], **Frank Hölzle**[2◉], **Omid Nikoubashman**[1] *

1 Department of Neuroradiology, University Hospital RWTH Aachen, Aachen, Germany, 2 Department of Oral and Maxillofacial Surgery, University Hospital RWTH Aachen, Aachen, Germany

◉ These authors contributed equally to this work.
* onikoubashman@ukaachen.de

## Abstract

### Objectives

The aim of this study was to demonstrate the efficacy and feasibility of treating patients with extracranial arteriovenous malformations (AVM) of the head and neck with interventional embolization followed by surgical resection.

### Methods

We reviewed the charts of all patients between 2012 and 2021 with extracranial AVM of the head and neck scheduled for interdisciplinary treatment according to University Hospital RWTH Aachen's protocol and conducted standardized interviews using a newly developed questionnaire. Interview results, as well as clinical examination and radiographic outcome results were analyzed to help determine the efficacy of our treatment approach.

### Results

We included 10 patients (8 female, 2 male), with a mean age of 33.5 (11–61) years who were scheduled for treatment of the AVM with interventional embolization followed by surgical resection. In 6 of the 10 patients (60%) the lesion was located in extracranial soft tissue only. In one patient (10%), the lesion was located in bone tissue only. A combined intraosseous and oral soft tissue lesion was seen in the remaining 3 patients (30%). Radiographic resolution was achieved in 62.5% of cases and a significant decrease of symptoms was identified (p = 0.002). None of the patients reported dissatisfaction and no major complications occurred.

### Conclusion

An interdisciplinary treatment approach combining neuroradiological interventions with surgical resection appears to be an effective treatment with an acceptable complication rate.

**Data Availability Statement:** All relevant data are within the manuscript and its Supporting Information files.

**Funding:** The authors received no specific funding for this work.

**Competing interests:** The authors have declared that no competing interests exist.

Patients treated according to our protocol showed a high satisfaction rate, regardless of the radiographic outcome. Standardized follow-up allows for early detection of recurrences and helps with subjective patient satisfaction.

## Introduction

Extracranial arteriovenous malformations (AVM) are rare vascular anomalies. These lesions are known to develop during early gestation and can be evident at birth [1]. They have a tendency to grow over time and become more prominent under the influence of trauma or hormonal changes during puberty or pregnancy [2].

According to the International Society of the Study of Vascular Anomalies (ISSVA), AVM are lesions of a benign nature that directly connect arteries with veins, and are lacking a capillary bed [3], leading to dilated and tortuous draining veins and deformities of the surrounding body region. Clinical presentation varies from slight discoloration and swelling of the skin to ulceration and gross disfigurement. Symptoms may include pain, a throbbing or pulsating sensation, discoloration, ulceration and even life-threatening haemorrhages [4]. In rare occasions, cardiovascular compromise with congestive heart failure has been reported [5]. All AVM symptoms potentially limit the patient's quality of life.

Established treatments for these lesions include conservative observation, transvascular embolization, percutaneous sclerotherapy and surgical excision. These strategies can be applied independently or in combination, though combined treatment approaches have been the treatment of choice in more recent years [6]. Regardless of the treatment method, high recurrence rates are still reported [1]. In clinical practice, treatment outcome analysis is considered complex due to prolonged intervention courses demanding long-term observation. In addition to objective data resulting from radiographic imaging, subjective patient satisfaction can also be taken into consideration using standardized questionnaires.

To reach our objectives, we analyzed interim and end-point results of an interdisciplinary neuroradiological and surgical treatment approach. We retrospectively extracted data of patients treated at our institution, analyzed their radiographic outcome, and conducted standardized interviews using the newly developed AQEM (Aachen Questionnaire for the Treatment of Extracranial Vascular Malformations) and the established UW-QOL (University of Washington Quality of Life Questionnaire).

## Materials and methods

We reviewed University Hospital RWTH Aachen files to identify all patients with extracranial vascular malformations seen in the Department of Neuroradiology between 2012 and 2021 after obtaining approval from the Ethics Committee at the RWTH Aachen Faculty of Medicine (Statement EK 023/21). All data were collected as clinical routine and were analyzed retrospectively and anonymously. The requirement for informed consent was waived by the ethics committee. Therefore, consent of individual participants was not obtained.

In total, 51 patients with extracranial vascular malformations were seen. These consisted of 18 venous malformations (VM), 5 lymphatic malformations (LM), 8 patients with mixed venous and lymphatic malformations (VM/LM), 4 capillary malformations (CM), and 16 patients with arteriovenous malformations (AVM).

We then extracted the files of all patients diagnosed and treated with extracranial AVM of the head and neck as seen in Fig 1. Digital subtraction angiography (DSA) and magnetic

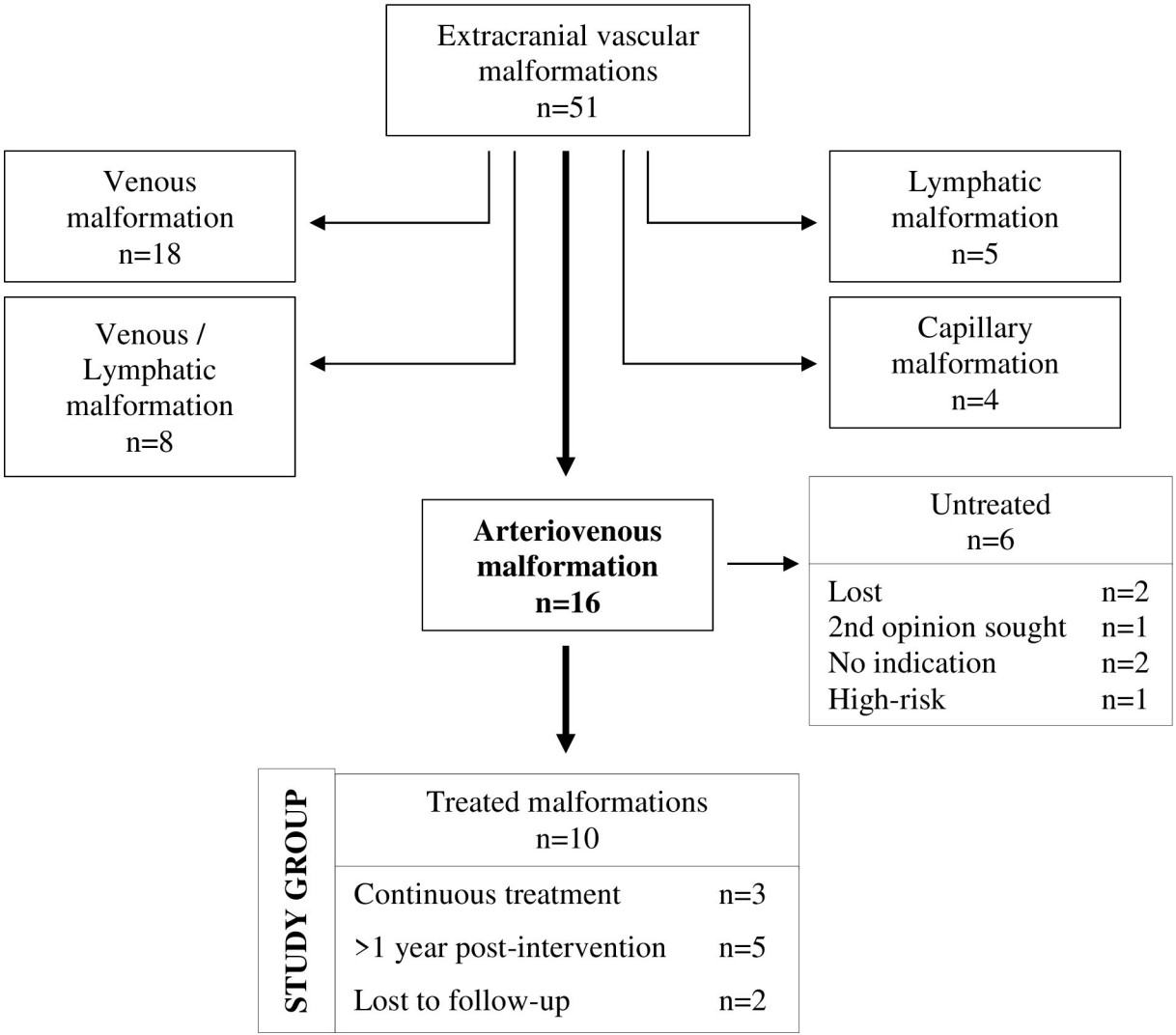

**Fig 1. Flow-chart of study group selection.**

resonance imaging (MRI), with demonstration of arterio-venous shunting, were used to confirm the diagnosis and collect baseline lesion volume. MRI was used for follow-up after the procedures. Additionally, two standardized questionnaires were administered to gather baseline information on symptoms, medical history, past treatment, and quality of life, as well as interim and follow-up results.

Data collection included sex, age at which symptoms were first noticed, age of first presentation at our institution, previous treatment received, lesion location and size, symptoms, course of treatment, total fluoroscopy time, post-interventional MRI, questionnaire results, documented complications, and follow-up time.

For radiographic analysis, proof of complete obliteration (>99% devascularization) of the nidus after neuroradiological intervention and, if applicable, surgical resection was considered as *cure*. Any presentation of residual arteriovenous shunting was considered as *incomplete treatment* and implicated continuous treatment.

The standardized interviews consisted of 1) the Aachen Questionnaire for the Treatment of Extracranial Vascular Malformations (AQEM) and 2) the German translation of the

University of Washington Quality of Life Questionnaire Version 4 (UW-QOL v4). Both questionnaires are provided in the supplement.

## Aachen questionnaire for the treatment of extracranial vascular malformations (AQEM)

This questionnaire was developed at our institution to standardize the initial diagnostic process, address the unique needs of each patient, and document follow-up procedures in a formalized manner. It is comprised of 29 questions divided into two sections. The first section is to be completed by a clinician during the initial consultation and consists of 23 questions on symptoms relating to the differentiation between the various types of vascular malformations. Medical history and treatment goals are also noted.

The second section, with five questions, is filled out by the patient each time they present at the clinic for consultation. Patients are asked about goals achieved, their reasons for seeking further treatment, any complication that arose during treatment, and their overall satisfaction.

## Washington quality of life questionnaire (UW-QOL v4)

The UW-QOL questionnaire consists of 12 single-question categories, each with three to six response options. Originally designed for patients with cancer of the head and neck, the questionnaire refers to health and physical life characteristics and to overall quality of life. Each response option is evenly scaled from 0 (worst) to 100 (best). Question categories cover pain levels, lesion appearance, patient activity and recreation, swallowing and chewing difficulty, speech, shoulder, taste, saliva, mood, and anxiety.

Results are divided into a physical subscale and a social-emotional subscale. According to the authors of the test, the mean normative reference scores are 95 ±10 for the physical subscale and 83 ±19 for the social-emotional subscale. Three global questions measure the patient's current quality of life as compared to their status before the onset of the disease, their overall health-related quality of life and their general quality of life, all on the same 0 to 100 scale. For this study we used the German translation of the 4th version of the UW-QOL questionnaire [7].

## Embolization procedure

Procedures were performed under either general anaesthesia or local anaesthesia, depending on patient condition. Access to the femoral artery was established using a 5 French or 6 French sheath. The nidus of the AVM was identified through selective angiography of the internal carotid artery (ICA) and external carotid artery (ECA). For endovascular access, a microcatheter (Rebar 18; Medtronic, Minneapolis, MN, USA) was advanced towards the nidus through the ECA. In case of vascular tortuosity, other microcatheters were used (SL10; Stryker, Kalamazoo, MI, USA or Marathon; Medtronic, Minneapolis, MN, USA) were used.

For endovascular particulate embolization, particles (Contour; Boston Scientific Corporation, Marlborough, MA, USA) were applied. For endovascular liquid embolization we used two agents (Onyx; ev3 Neurovascular, Irvine, CA, USA and Glubran 2; GEM SRL, Viareggio, Italy. For percutaneous interstitial embolization (sclerotherapy), Bleomycin mixed with contrast medium (50:50) was applied using 26–30 Gauge needles under fluoroscopic guidance.

Post-treatment, catheter angiography and MR angiography were performed to confirm successful elimination of the AVM nidus.

## Surgical procedure

Surgical intervention was performed by the Department of Oral and Maxillofacial Surgery at University Hospital RWTH Aachen. For soft tissue and intramuscular AVM, excision or extirpation was performed after preoperative embolization. Legation of afferent vessels followed by excision of the AVM without visible residual masses was achieved for all patients. Particular attention was paid to preserving nerve structures. For cosmetic reconstruction, a stalked skin flap plastic or a cheek rotation skin flap was used, as needed.

For intraosseous lesions, partial resection of the involved bone or hemimandibulectomy was performed. After submarginal incision, the platysma was exposed and surged. The submandibular gland was extirpated after legation of supplying vessels. Exposure of the carotid sheath improved line of sight to vascular structures for subsequent anastomosis. After temporarily tying off the lingual artery, the affected area of the mandible was detached and sent to the Institute of Pathology for examination. Replacement was provided with an osteocutaneous or musculocutaneous transplantation of iliac crest or fibula. To provide accurately fitted transplants, digital construction of templates with the help of computer software (ProPlan CMF; DePuy Synthes, West Chester, PA, USA) was used. The templates were intraoperatively fitted to the harvest bone site by screws, providing the exact dimension of the needed transplant. Making use of the previously exposed vascular structures, the transplants were attached with osteosynthesis sheets and bloodflow was restored. Significant effort was made to minimize ischemic time and ensure adequate perfusion of the transplant bone marrow post-implementation.

Patients with hemimendibulectomy or bone resection had their airways secured with temporary tracheotomy; in all other cases, endotracheal intubation was performed. Post-surgical patients were placed in an intensive or intermediate care unit and monitored for immediate surgical complications or haemorrhage. Patient condition and vital signs were monitored for a minimum of 24 hours. Clinical follow-up consultations were established in intervals of three to six weeks after discharge from our hospital.

## Statistical analysis

For statistical analysis we used cross-table, Pearson and Spearman calculations. All statistical analyses were performed with a software platform (SPSS Version 27.0; IBM, Armonk, NY, USA).

## Results

Of the 51 patients with vascular malformations (AVM, VM, LM, Capillary Malformation) who were seen by our Neuroradiology Department between 2012 and 2021, 16 were diagnosed with an extracranial AVM of the head and neck region. Ten of those patients were scheduled for interdisciplinary treatment which was introduced in 2012 in collaboration with the Department of Oral and Maxillofacial Surgery, and thus comprised our study group.

There were 8 females and 2 males, with a mean age of 33.5 ±16.8 years (range 11–61 years) at first presentation. Four patients had not undergone any treatment while the other six patients had previously received treatment, including surgical interventions, embolization procedures and laser or cryotherapy. A comprehensive overview of the study group's demography is provided in Table 1.

In 6 patients (60%), the AVM lesion was located in soft tissue only. In one patient (10%), the lesion was located in bone tissue only. The remaining 3 patients (30%) had a combined intraosseous and soft tissue lesion.

In 3 of the 10 patients (30%), the AVM was located on the nose, in 2 (20%) the lesion was in the cheek and lip, and 5 patients (50%) had intraoral involvement. All combined intraosseous

and soft tissue AVM were located intraorally. The mean volume of the lesions before treatment was 70.8 ±63.1 ml (10.1–167.4 ml).

The mean age, at which symptoms were first noticed was 19.9 ±18.7 years (1–57 years). All 10 patients (100%) experienced swelling. Discoloration of the skin occurred in 50% (5/10) of cases, bleeding in 30% (3/10), and a pulsating sensation in 20% (2/10). The lesion volume had progressed over time in all patients (10/10). Nine out of ten patients responded to the question about the reasons they sought treatment as follows: Appearance in 66.7% (6/9); difficulty in chewing in 44.4% (4/9); paresthesia in 44.4% (4/9); pain in 33.3% (3/9); reduced mental well-being in 22.2% (2/9); bleeding in 22.2% (2/9); difficulties in swallowing in 11.1% (1/9); and difficulties with speech in 11.1% (1/9).

All ten patients underwent endovascular embolization. In 6 patients, endovascular treatment was comprised solely of particle embolization. One patient was treated with a combination of particle and coil embolization, and a second patient received a combination of particle embolization and sclerotherapy using Bleomycin. A third patient received liquid embolization only using both Onyx and Glubran, while one other patient received combined particle, Bleomycin, Aethoxysklerol and Glubran embolization. Mean accumulated fluoroscopy time during the embolization procedures was 101.7 ±43.5 minutes (29.9–192.7 minutes).

Nine patients underwent surgery after the endovascular treatment; one of the ten patients refused surgical resection. The mean time between embolization procedure and surgical intervention was 3.3 ±4.1days (1–14 days).

After treatment, eight of the ten patients underwent follow-up imaging and radiographic evaluation; one patient had permanent residency out of country, and one patient refused. Re-evaluation of the lesions via MRI or angiography showed complete obliteration of the nidus, considered as *cure* in five of the eight patients (62.5%). A mean time of 1066 ±693.8 days (376–2004 days) had passed since treatment. The three remaining patients presented with residual niduses and were under continuous treatment at the time of analysis.

Patients who had finished treatment were hospitalized for a median of 12 (Interquartile range 7–47) days. All but two patients experienced minor adverse effects, limited to the region of former lesion. Minor side effects included paresthesia, swelling and pain. In all cases, side effects were managed conservatively. None of the patients experienced major complications, in accordance with the Society of Interventional Radiology (SIR) classification [8].

In the AQEM questionnaire, 77.7% (7/9) of patients were very satisfied or mostly satisfied with the course of treatment. Two patients were undecided about their satisfaction with treatment. None of the patients showed dissatisfaction. When looking at the subgroups, 80% (4/5) of radiographically *cured* patients, and 66.7% (2/3) of radiographically *incomplete treated* patients reported to be very satisfied or mostly satisfied.

**Table 1. Demographic characteristics of study group.**

| Patients in study group | 10 |
| --- | --- |
| Male / Female | 2 / 8 |
| Mean age (Range) in years | 33.5 (11–61) |
| Previously treated patients | 6 |
| Previous treatment *<br>  Surgery<br>  Embolization<br>  Laser<br>  Cryotherapy | <br>3<br>4<br>1<br>1 |
| AVM intramuscular / intraosseous | 7 / 3 |

* Multiple treatment may apply.

In the UW-QOL v4 questionnaire, patients showed a mean score of 89.3 ±11.8 (61.7–100) in the physical subscale (the mean normative reference score is 95 ±10). The mean score in the social-emotional subscale for our patient collective was 82.5 ±17.4 (50–100) (the mean normative reference score for this subscale is 83 ±19). Two patients did not participate in the interview.

When asked about their treatment goals, patients reported a mean of 2.7 ±1.8 (1–6) symptoms they desired to improve before the initial treatment at our institution. After initiation of the treatment, patients reported a mean of 1.1 ±1.1 (0–3) symptoms, showing a significant decrease in symptoms over the treatment period (p = 0.002).

Complete follow-up data sets (neuroradiological imaging and interviews) were collected from 7 of the 10 patients; one patient had permanent residency abroad and did not participate in follow-up imaging or interviews. Two patients participated in the interview only and did not undergo follow-up imaging. A comprehensive overview of the treatment and outcome in the study group is provided in Table 2.

## Discussion

Interventional embolization and surgical resection are both established approaches for definite treatment of extracranial AVM of the head and neck. However, a recent meta-analysis of published studies involving interventional treatment of this condition showed that there is currently no treatment or reporting standard for extracranial AVM of the head and neck, and that the methodological quality of available publications is too heterogeneous to draw conclusions on optimal treatment strategies [9]. To cite just one example, some authors have termed devascularization rates of 90% as "cure", while others have presented treatment results without any radiographic modality for follow-up [10–12].

To the best of our knowledge, no prospective clinical study has been published on this disease. Minimum requirements and standards were recommended for all clinical studies on the treatment of extracranial AVM of the head and neck to ensure sufficient data for the development of an evidence-based treatment strategy. These include: 1) a dynamic and/or angiographic imaging modality confirming the AVM and distinguishing it from other vascular entities; 2) a detailed description of the therapeutic management; 3) the definition of a devascularization of greater than 99% for complete resolution; 4) a follow-up time of at least one year with post-treatment radiographic imaging. Additionally, we hypothesize that questionnaires may also be a helpful tool to evaluate treatment strategies and measure patient satisfaction.

In line with recent recommendations, treatment protocol for extracranial AVM at our institution consists of a combination of endovascular embolization followed by surgical resection [6]. In this study we retrospectively analyzed our results using the above-mentioned reporting standards. Radiographic resolution (*cure*) was achieved in 62.5% of patients with a mean follow-up period of 1066 days. In the questionnaires we used, 80% of cured patients were satisfied with treatment. Significantly fewer symptoms were reported after treatment. No major complications occurred in our study group. Minor sensory deficits and aesthetic affectation were reported by the patients during follow-up consultation and need to be further addressed for complete patient satisfaction. Overall, the rate and extent of complications has been shown to be acceptable to both patients and treating physicians.

Our results appear compatible with other publications. In studies where endovascular therapy was the sole treatment modality, or where endovascular therapy was combined with surgery, rates for "success" or "improved condition" ranged between 44% and 100% [10,11,13–31]. We believe that the heterogeneity of reporting standards used by the studies outlined in

**Table 2. Comprehensive overview of results in the study group.**

| Case | Sex | Age | Previous treatment | Location | | Initial Size [mm] | Treatment Neuroradiology | Treatment Surgery | Flouroscopy time [min] | Outcome | Adverse effects | Radiographic Follow-Up [days] |
|---|---|---|---|---|---|---|---|---|---|---|---|---|
| 1 | m | 47 | None | Nose | Soft tissue | 35x20x15 | Particles/ Bleomycin | Excision | 125.7 | Cure | Paraesthesia | 2004 |
| 2 | f | 11 | Embolization | Floor of mouth | Soft tissue / Intraosseous | 79x36x51 | Particles | Hemimandibulectomy / Osteocutaneous fibula transplant | 114.5 | | Failed to contact | |
| 3 | f | 12 | None | Mandible, Floor of mouth | Soft tissue/ Intraosseous | 71x21x40 | Particles | Excision | 62.5 | Cure | Paraesthesia | 1456 |
| 4 | f | 31 | Excision | Cheek, Lip | Soft tissue | 32x15x21 | Particles | Excision | 29.9 | No relaps in questionnaire | None | None |
| 5 | f | 30 | Embolization/ Excision | Mandible, Floor of mouth | Soft tissue/ Intraosseous | 42x73x53 | Particles | Excision / ALT-Transplant | 90.9 | Cure | Praesthesia, Bleeding | 1075 |
| 6 | f | 42 | Laser/Excision | Cheeck, Lip, Intraoral | Soft tissue | lower lip:18x11x27 upper lip:15x7x12 tongue: 15x50x22 | Particles/ Bleomycin/ Aethoxysklerol/ Glubran | Refused | 88.6 | Ongoing treatment | Pain, Discoloration | Current treatment |
| 7 | f | 61 | Biopsy | Nose | Soft tissue | 44x63x14 | Particles | Excision / Stalked skin flap | 192.7 | Ongoing treatment | Aesthetic | Current treatment |
| 8 | m | 51 | Laser/ Cryotherapy | Nose | Soft tissue | 41x28x25 | Particles/Coils | Excision / Stalked skin flap / Cheek rotation skin flap | 80.9 | Ongoing treatment | None | Current treatment |
| 9 | f | 18 | Embolization | Mandible, Floor of mouth | Intraosseous | 62x50x54 | Particles | Hemimandibulectomy / Osteocutaneous fibula transplant | 106.0 | Cure | Abscess | 420 |
| 10 | f | 32 | None | Mandible, Floor of mouth | Soft tissue | 48x36x36 | Onyx/Glubran | Extirpation | 125.2 | Cure | Paraesthesia, Pain, Swelling | 376 |

f = female, m = male, ALT-Transplant = anterolateral thigh transplant.

other publications is too large to allow definitive conclusions on therapeutic efficacy. Objective definitions or reproducible proxies for successful treatment are often missing and follow-up procedures are inconsistent. Moreover, useful outcome measurement tools for extracranial AVM that reflect all aspects have yet to be developed [32].

In coronary interventions, prolonged fluoroscopy has been shown to indicate higher procedure complexity [33]. Voluminous intracranial AVM, too, could assumptively have a higher risk for morbidity and mortality [34]. However for extracranial AVM of the head and neck, we did not identify a significant correlation between total fluoroscopy time and initial volume of the lesion (p = 0.814). Fluoroscopy time and radiographic result did not have a significant correlation either (p = 0.895). To illustrate this, in case #1, where the lesion volume was 10.5 ml and fluoroscopy time during intervention 125.7 minutes, the lesion was completely resolved on MRI after one year follow-up. In case #5, with an initial lesion volume of 162.5 ml and a fluoroscopy time of 90.9 minutes, the lesion was also completely resolved. Therefore, fluoroscopy time can neither be used as a surrogate marker for lesion volume or complexity, nor as a predictor for the result. Scatter diagrams for both correlations are provided in the supplement.

Subjective patient satisfaction surveys have been used to measure outcome alongside other quality of treatment indicators [35]. However, their validity has been controversially discussed [36]. In our interviews, 80% of cured patients (>99% devascularization in radiographic imaging) reported to be very satisfied or mostly satisfied with the treatment at our institution. This is supported by the social emotional subscale score in the UW-QOL with a mean patient score of 82.5 points, which is within the range of the healthy population. With regard to this, patients experience less of a deficit than radiographic outcomes suggest. The UW-QOL may not be ideal for our study group, however, since it was originally designed for patients with cancer of the head and neck [37]. Subjective patient satisfaction may be one indicator for treatment that needs more attention for outcome analysis in addition to radiographic studies.

With the development of the AQEM questionnaire as patient reported outcome measure, we have established a tool to closely monitor and adjust treatment strategies as required. Future prospective trials and long-term observation studies ought to be conducted to verify this hypothesis.

## Conclusion

Our results indicate that our protocol to treat extracranial AVM of the head and neck using preoperative embolization and consecutive surgical resection is both effective and feasible and lead to high patient satisfaction. However, more published articles using appropriate reporting standards and prospective clinical studies are needed to identify comprehensive outcome proxies and develop optimal treatment protocols for these lesions, which are challenging to treat and considerably undermine patients' quality of life.

## Supporting information

**S1 Supplement. AQEM questionnaire.**
(PDF)

**S2 Supplement. UW-QOL v4 questionnaire.**
(PDF)

**S3 Supplement. Scatter diagram for correlation between fluoroscopy time and initial lesion size.**
(DOCX)

**S4 Supplement. Scatter diagram for correlation between fluoroscopy time and radiographic outcome.**
(DOCX)

**S5 Supplement. Full data set.**
(XLSX)

## Acknowledgments

We would like to express our special thanks of gratitude to our friend, Sara Fraser, who has supported our work with her language skills and proofread the manuscript.

## Author Contributions

**Conceptualization:** Daniel Lilje, Martin Wiesmann, Frank Hölzle, Omid Nikoubashman.

**Data curation:** Daniel Lilje, Dimah Hasan, Omid Nikoubashman.

**Formal analysis:** Daniel Lilje.

**Investigation:** Daniel Lilje, Dimah Hasan, Hani Ridwan, Frank Hölzle, Omid Nikoubashman.

**Methodology:** Daniel Lilje, Martin Wiesmann.

**Project administration:** Omid Nikoubashman.

**Resources:** Frank Hölzle.

**Supervision:** Martin Wiesmann, Omid Nikoubashman.

**Validation:** Hani Ridwan.

**Writing – original draft:** Daniel Lilje.

**Writing – review & editing:** Daniel Lilje, Martin Wiesmann, Dimah Hasan, Hani Ridwan, Frank Hölzle, Omid Nikoubashman.

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
