## [Decision Letter · Decision Letter 0]

28 Jun 2022

PONE-D-22-08407Interventional embolization combined with surgical resection for treatment of extracranial AVM of the head and neck: A monocenter intention-to-treat analysisPLOS ONE

Dear Dr. Nikoubashman, please note that the previously assigned Editor has cancelled his cooperation with Plos One recently. I have stepped in here yesterday, so please accept our apologies for any delay with this review process.

Thank you for submitting your manuscript to PLOS ONE. After careful consideration, we feel that it has merit but does not fully meet PLOS ONE’s publication criteria as it currently stands. Therefore, we invite you to submit a revised version of the manuscript that addresses the points raised during the review process.

Having intensively reviewed your revised draft, our external reviewers differed to some extent with their final recommendations. Additionally, I have double checked your submitted version, to come to a more balanced decision (see R #3). All in all, I am convinced that your re-revised paper will be worth following, even if your current version still would benefit from thorough re-edits and some language polishing. Thus, I would like to encourage you to provide a thorough (in terms of language, reviewers' constructive criticism, content, generalizable outcome, and/or Authors' Guidelines) revision in order to avoid an iterative and lengthy review process and facilitate a smooth publication process.

We look forward to receiving your revised manuscript.

Kind regards,

Andrej M Kielbassa, Prof. Dr. med. dent. Dr. h. c.

Academic Editor

PLOS ONE

Journal Requirements:

Reviewers' comments:

Reviewer's Responses to Questions

**Comments to the Author**

1. Is the manuscript technically sound, and do the data support the conclusions?

Reviewer #1: Yes

Reviewer #2: Yes

Reviewer #3: No

2. Has the statistical analysis been performed appropriately and rigorously? 

Reviewer #1: Yes

Reviewer #2: Yes

Reviewer #3: Yes

3. Have the authors made all data underlying the findings in their manuscript fully available?

Reviewer #1: Yes

Reviewer #2: Yes

Reviewer #3: Yes

4. Is the manuscript presented in an intelligible fashion and written in standard English?

Reviewer #1: Yes

Reviewer #2: Yes

Reviewer #3: Yes

5. Review Comments to the Author

Reviewer #1: The authors should be complimented for their manuscript. It gives important insights into the treatment of a very rare disease and the patient number is for such a disease reasonable with 10 patients. Their conclusion is warranted by their experience.

However, there are a few minor points to address:

Title:

As this is a retrospective analysis, how can it be an intention to treat analysis? In my opinion this is only possible in prospective studies.

Abstract:

“Patients treated according to our protocol show a high satisfaction rate, regardless of the radiographic outcome.” Please use past tense (showed)

Advances in knowledge:

In my opinion the statement is too strong, please consider changing it to “..might be an effective..”

General comment:

Sometimes the mean is used and sometimes the median. It should be checked if in cases of mean normality is given.

The results are presented in a transparent way and the conclusions are warranted by the results. The discussion is well written and all relevant literature was cited.

Reviewer #2: The paper by Lije et al. analyzed the efficacy and feasibility of combined (interventional embolization plus surgery) for the treatment of extracranial AVMs by analyzing objective parameters (i.e., radiographic proof of obliteration, adverse effects) as well as subjective parameters (AQUEM questionnaire, UW-QOL v4 questionnaire). Therefore, of all patients with vascular malformations seen by the Institution's Department of Neuroradiology between 2012 and 2021, those with an extracranial AVM scheduled for interdisciplinary treatment were retrospectively analyzed.

The article shows all essential data regarding patient, intervention, and outcome. It is well structured in its methodology and thereby provides comprehensible results. The results are well defined, and the findings are well embedded and addressed in the discussion. Although the analyzed cases are relatively low, the general message that combined treatment is practical and feasible in selected patients is well supported.

Two statements seem unclear and could be enhanced with minor revisions:

- In line 169, you state that surgery was performed within seven days of the embolization procedure; however, in line 238, the range for the time between procedure and surgery is 1-14d.

- in your results (line 242), you state that five of eight patients were considered cured, but in supplement four, there seem to be four in the whole group and four in the incomplete group (Misplacement of Case 9?); Does that influence the statistical calculations as well?

I am happy to look at the revised version and congratulate you on the excellent work.

Reviewer #3: This paper has been well elaborated, no doubt. Notwithstanding, the Authors have missed to strictly follow the Journal style, see Guidelines, and consult some recently published Plos One papers.

- With your Abstract section, please please provide as much information as possible within the allowed 300-word limit. Add p values with your most prominent results. Style must be "(p < 0.001)", or "(p = 0.012)".

- Reference style has to be adapted to the Journal guidelines. "(...) and can be evident at birth.(1)" must read "(...) and can be evident at birth¹."

- With ALL materials (including chemicals) and methodologies (including statistical software), please use general names with your text, followed by (brand name; manufacturer, city, St[ate, abbreviated, if US], country) in parentheses. Stick to semicolon. Revise thoroughly.

- With your Conclusions, please stick exclusively to your revised aims. Do not simply repeat your results here. Do not speculate on future studies. Instead, provide a reasonable and generalizable extension of your outcome.

- Please revise your reference list for uniform formatting. Style would be "Kasraei S, Sami L, Hendi S, Alikhani MY, Rezaei-Soufi L, Khamverdi Z. Antibacterial properties of composite resins incorporating silver and zinc oxide nanoparticles on Streptococcus mutans and Lactobacillus. Restor Dent Endod. 2014; 39(2): 109–114. https://doi.org/10.5395/rde.2014.39.2.109 PMID: 24790923" Provide doi and PMID numbers.

6. PLOS authors have the option to publish the peer review history of their article (what does this mean?). If published, this will include your full peer review and any attached files.

Reviewer #1: No

Reviewer #2: No

Reviewer #3: No

---

## [Author Response · Author response to Decision Letter 0]

12 Jul 2022

Dear Editors, 

Dear reviewers,

We would like to thank you for the thoughtful and favorable reviews and the opportunity to revise our manuscript. We believe that our manuscript has benefitted considerably from the remarks, and we hope that we addressed the mentioned issues to your satisfaction. 

Below you can find our responses to the reviewer’s comments in red. Please also find attached our revised manuscript in two versions: with and without tracked changes.

Additionally, we have added the full data set of our study as supplement S5 file.

We are happy to hear from you if you have any other question.

With our highest regards,

The authors

Reviewer #1: […]

Title: As this is a retrospective analysis, how can it be an intention to treat analysis? In my opinion this is only possible in prospective studies.

We would like to thank the reviewers as he/she accurately points out that an intention-to-treat analysis is defined as a prospective study design. We have made according changes in the title:

Interventional embolization combined with surgical resection for treatment of extracranial AVM of the head and neck: A monocentric retrospective analysis

Abstract:

“Patients treated according to our protocol show a high satisfaction rate, regardless of the radiographic outcome.” Please use past tense (showed)

We would like to give thanks to the reviewer for the correction of this mistake. We changed the sentence into past tense.

Patients treated according to our protocol showed a high satisfaction rate, regardless of the radiographic outcome.

(Clean copy lines 52-53)

Advances in knowledge:

In my opinion the statement is too strong, please consider changing it to “..might be an effective..”

In consideration of the small sample size and the rarity of the condition, the reviewer is correct that a less strong wording might be helpful to stimulate a discussion.

The results of this study indicate that embolization followed by surgical resection might be an effective treatment for extracranial AVM of the head and neck and is accompanied by high patient satisfaction.

(Clean copy lines 57-59)

[…]

Sometimes the mean is used and sometimes the median. It should be checked if in cases of mean normality is given.

We would like to thank the reviewer for the helpful remark to check the usage of the mean and median. We used the mean for statistical analysis wherever the distribution of data was expected to be symmetrical (e.g. patients’ age, lesion volume, accumulated fluoroscopy time), or if reference scores where provided as such (e.g. in the UW-QOL v4 questionnaire). The median was used only in reference to the hospitalization time, where outliers could be identified as shown in the comprehensive overview of the study group (Table 2). In careful attention of these statistical aspects, we deem the utilization to be appropriate.

Reviewer #2: […]

Two statements seem unclear and could be enhanced with minor revisions:

- In line 169, you state that surgery was performed within seven days of the embolization procedure; however, in line 238, the range for the time between procedure and surgery is 1-14d.

The reviewer is correct to point out the mismatching specification of time between embolization procedure and surgery. The passage on the surgical procedure merely tries to give an overview on the course of action of the procedure. The timeframe of seven days was communicated between the neuroradiological and surgical department to facilitate smooth workflow. To prevent the number in this section to be confused as part of the results, we erased the section.

Surgical intervention was performed by the Department of Oral and Maxillofacial Surgery at University Hospital RWTH Aachen.

(Clean copy lines 165-166)

- in your results (line 242), you state that five of eight patients were considered cured, but in supplement four, there seem to be four in the whole group and four in the incomplete group (Misplacement of Case 9?); Does that influence the statistical calculations as well?

We would like to thank the reviewer for this very important note. There was indeed a mistake in the scatter diagram and case #9 was wrongfully placed in the “incomplete” group. This has no effect on the calculations as they were performed independently. The revised copy of the scatter diagrams can be found in file S3 Supplement 3 and file S4 Supplement 4.

Supplement 3. Scatter diagram for correlation between fluoroscopy time and initial lesion size (p=0.814).

Supplement 4. Scatter diagram for correlation between fluoroscopy time and radiographic outcome (p=0.895).

Reviewer #3:

- With your Abstract section, please please provide as much information as possible within the allowed 300-word limit. Add p values with your most prominent results. Style must be "(p < 0.001)", or "(p = 0.012)".

We are happy for the reviewer to bring this aspect to our attention. As most of the findings are of descriptive nature and significance of comparison had lower priority in the analysis, the highlights of the results are listed in the abstract. We have added the significant decrease of symptoms (p=0.002) as another prominent finding.

Radiographic resolution was achieved in 62.5% of cases and a significant decrease of symptoms was identified (p=0.002). None of the patients reported dissatisfaction and no major complications occurred.

(Clean copy lines 45-47)

- Reference style has to be adapted to the Journal guidelines. "(...) and can be evident at birth.(1)" must read "(...) and can be evident at birth¹."

We would like to thank the reviewer for this remark. All citations have been adapted accordingly.

- With ALL materials (including chemicals) and methodologies (including statistical software), please use general names with your text, followed by (brand name; manufacturer, city, St[ate, abbreviated, if US], country) in parentheses. Stick to semicolon. Revise thoroughly.

The reviewer brings up a very good point that interrupted the flow of reading significantly. We have corrected the sections according to the recommendations.

- With your Conclusions, please stick exclusively to your revised aims. Do not simply repeat your results here. Do not speculate on future studies. Instead, provide a reasonable and generalizable extension of your outcome.

This remark by the reviewer was very helpful for the authors to reconsider the key findings of the article and sparked a fruitful discussion. With the presented treatment protocol and its results, we are certain to only show one possible way of managing this rare condition. In view of existing literature, the conclusion of the article may best be as cautious as outlined in the current manuscript to give room and motivation for further studies. However, if changes in the regard to this aspect are needed, we are happy to adapt the manuscript.

- Please revise your reference list for uniform formatting. Style would be "Kasraei S, Sami L, Hendi S, Alikhani MY, Rezaei-Soufi L, Khamverdi Z. Antibacterial properties of composite resins incorporating silver and zinc oxide nanoparticles on Streptococcus mutans and Lactobacillus. Restor Dent Endod. 2014; 39(2): 109–114. https://doi.org/10.5395/rde.2014.39.2.109 PMID: 24790923" Provide doi and PMID numbers.

The reviewer is correct to bring this point to our attention. We have made changes in the bibliography according to the recommendations in the PLOS submission guidelines. We provided the PMID if DOI could not be obtained.

---

## [Decision Letter · Decision Letter 1]

1 Aug 2022

Interventional embolization combined with surgical resection for treatment of extracranial AVM of the head and neck: A monocentric retrospective analysis

PONE-D-22-08407R1

Dear Dr. Nikoubashman,

We’re pleased to inform you that your manuscript has been judged scientifically suitable for publication and will be formally accepted for publication once it meets all outstanding technical requirements. Congratulations, and stay healthy!

Kind regards,

Andrej M Kielbassa, Prof. Dr. med. dent. Dr. h. c.

Kind regards,

Andrej M Kielbassa

Academic Editor

PLOS ONE

Additional Editor Comments (optional):

Reviewers' comments:

Reviewer's Responses to Questions

**Comments to the Author**

1. If the authors have adequately addressed your comments raised in a previous round of review and you feel that this manuscript is now acceptable for publication, you may indicate that here to bypass the “Comments to the Author” section, enter your conflict of interest statement in the “Confidential to Editor” section, and submit your "Accept" recommendation.

Reviewer #1: All comments have been addressed

Reviewer #2: All comments have been addressed

Reviewer #3: All comments have been addressed

2. Is the manuscript technically sound, and do the data support the conclusions?

Reviewer #1: Yes

Reviewer #2: Yes

Reviewer #3: Yes

3. Has the statistical analysis been performed appropriately and rigorously? 

Reviewer #1: Yes

Reviewer #2: Yes

Reviewer #3: Yes

4. Have the authors made all data underlying the findings in their manuscript fully available?

Reviewer #1: Yes

Reviewer #2: Yes

Reviewer #3: Yes

5. Is the manuscript presented in an intelligible fashion and written in standard English?

Reviewer #1: Yes

Reviewer #2: Yes

Reviewer #3: Yes

6. Review Comments to the Author

Reviewer #1: All comments wäre adressed.

Reviewer #2: In the revised version of this manuscript, the reviewer's comments have been integrated in a valuable way, and the manuscript has profited substantially.

It is still a tiny series but a thorough workup and evaluation of the cases, providing an excellent base for further studies and fruitful discussion on that topic.

I congratulate the authors for their work.

Reviewer #3: This revised and re-submitted draft has been considerably improved, and all comments have been adequately addressed. This manuscript is considered ready to proceed.

7. PLOS authors have the option to publish the peer review history of their article (what does this mean?). If published, this will include your full peer review and any attached files.

Reviewer #1: No

Reviewer #2: No

Reviewer #3: No

---

## [Editor Report · Acceptance letter]

22 Aug 2022

PONE-D-22-08407R1 

Interventional embolization combined with surgical resection for treatment of extracranial AVM of the head and neck: A monocentric retrospective analysis 

Dear Dr. Nikoubashman:

I'm pleased to inform you that your manuscript has been deemed suitable for publication in PLOS ONE. Congratulations! Your manuscript is now with our production department. 

Kind regards, 

on behalf of

Prof. Dr. med. dent. Dr. h. c. Andrej M Kielbassa 

Academic Editor

PLOS ONE